# Historical Osteopathic Principles and Practices in Contemporary Care: An Anthropological Perspective to Foster Evidence-Informed and Culturally Sensitive Patient-Centered Care: A Commentary

**DOI:** 10.3390/healthcare11010010

**Published:** 2022-12-21

**Authors:** Rafael Zegarra-Parodi, Francesca Baroni, Christian Lunghi, David Dupuis

**Affiliations:** 1A.T. Still Research Institute, A.T. Still University, Kirksville, MO 63501, USA; 2BMS Formation, 75116 Paris, France; 3Centre Européen d’Enseignement Supérieur de l’Ostéopathie (CEESO) Paris, 93200 Saint-Denis, France; 4Osteopatia Lunghi-Baroni Private Practice, 00146 Rome, Italy; 5Institut National de la Santé et de la Recherche Médicale, Institut de Recherche Interdisciplinaire sur les Enjeux Sociaux (INSERM/IRIS), 93300 Aubervilliers, France

**Keywords:** anthropology, medical, complementary therapies, culturally sensitive care, health belief model, historical osteopathic practices, historical osteopathic principles, manipulation, osteopathic, patient-centered care, medicine, traditional, Western healthcare

## Abstract

Historical osteopathic principles and practices (OPP)—considering the patient as a dynamic interaction of the body, mind, and spirit and incorporating the body’s self-healing ability into care—are inherited from traditional/complementary and alternative (CAM) principles. Both concepts are familiar to contemporary osteopathic practitioners, but their incorporation into healthcare for evidence-informed, patient-centered care (PCC) remains unclear. Further, a polarity exists in the osteopathic profession between a ‘traditional-minded’ group following historical OPP despite evidence against those models and an ‘evidence-minded’ group following the current available evidence for common patient complaints. By shifting professional practices towards evidence-based practices for manual therapy in line with the Western dominant biomedical paradigm, the latter group is challenging the osteopathic professional identity. To alleviate this polarity, we would like to refocus on patient values and expectations, highlighting cultural diversity from an anthropological perspective. Increasing an awareness of diverse sociocultural health assumptions may foster culturally sensitive PCC, especially when including non-Western sociocultural belief systems of health into that person-centered care. Therefore, the current medical anthropological perspective on the legacy of traditional/CAM principles in historical OPP is offered to advance the osteopathic profession by promoting ethical, culturally sensitive, and evidence-informed PCC in a Western secular environment. Such inclusive approaches are likely to meet patients’ values and expectations, whether informed by Western or non-Western sociocultural beliefs, and improve their satisfaction and clinical outcomes.

## 1. Introduction

Dr. Andrew Taylor Still, a Doctor of Medicine (MD) and Doctor of Osteopathy (DO), was the founder of the osteopathic profession and a visionary who saw the cross-cultural potential of healthcare while living and working on the rural American frontier in the 19th century. As an American physician who interacted with Native American populations, he became aware of their healing principles and created osteopathy as a unique healing system. Relying on historical osteopathic principles and practices (OPP), osteopathy combined spirit-based healing wisdom inherited from traditional/complementary and alternative medicine (CAM) and Western scientific-based healing wisdom [1].

The development, recognition, and regulation of the osteopathic profession throughout the world has been subject to the opposing viewpoints of an appreciation from the public and challenges from academics, especially for OPP that were contrary to the dominant Western biomedical system [2]. For example, manual skills historically introduced by the osteopathic profession, such as cranial and visceral techniques, are popular among specific patient populations, but the current evaluation of those techniques has failed to provide clinically meaningful or academically acceptable data [3,4]. Such manual interventions raise ethical concerns for patients when the underlying models are considered pseudoscience [5] and represent a clear academic challenge for educators.

The osteopathic profession is not the only one with a long history of internal conflict [6]; however, the current pressures of evidence-based healthcare have worsened the opposing views of historical OPP [7]. Presently, there is a ‘traditional-minded’ group that follows historical OPP concepts despite evidence against those models and an ‘evidence-minded’ group that focuses on contemporary and evidence-based practice to treat musculoskeletal (MSK) problems. This duality of osteopathic practices, one led by historical OPP and the other by evidence, is clearly problematic for the profession, especially given the general acceptance of evidence-based practice for the ethical provision of care in Western industrialized societies [7]. By combining Western and non-Western principles to guide care, Still’s seminal cross-cultural perspective for healthcare challenges many modern osteopathic practitioners who believe the historical aspects of osteopathy mislead current professional practice and should, thus, be discarded [8].

To move beyond these conflicting professional values, the current essay proposes to refocus on patient values and expectations. In conjunction with practitioner experience and the best available evidence, the values and expectations form the three pillars of evidence-based practice for ethical and inclusive healthcare [9]. Because medicine constructs the body and ascribes meaning, it should be considered a mechanism of sociocultural systems, where the conception of what constitutes health and what warrants intervention changes according to the sociocultural environment [10]. For example, the way individuals perceive, express, and control pain is a learned behavior specific to each culture [11]. Defined as a set of rules and standards developed over time and shared by members of a particular society, culture is the hallmark of anthropology, which studies human behavior and culture [11]. From a clinical perspective, culturally sensitive patient-centered care (PCC) involves modifiable behaviors and attitudes by practitioners that address patients’ values and expectations in a healthcare environment that culturally diverse patients will recognize as respectful of their culture, enabling them to feel comfortable with and respected by their practitioner [12]. Patient-centered care is defined by its four core elements: considering each patient as a person, using a biopsychosocial perspective, sharing power and responsibility, and establishing the therapeutic alliance [13]. Importantly, this type of care is based on the views of culturally diverse patients rather than the views of healthcare professionals [12].

Rooted in Western and non-Western healing principles, it is likely that the osteopathic profession initially developed distinctive practices that attracted a broader spectrum of patients than other professions that relied only on Western healing principles. The early development of the osteopathic profession may be explained by the inclusion of values and expectations of culturally diverse patients into healthcare. To our knowledge, the investigation of historical OPP has not included an anthropological perspective of the values and expectations of non-Western patients within a Western evidence-based environment. Such cross-cultural considerations may help practitioners move beyond Western biomedical ethnocentrism and promote culturally sensitive PCC for patients with traditional/CAM belief systems based in historical OPP. Seeing and interpreting the world cross-culturally is certainly novel for many practitioners, but doing so promotes inclusive approaches and shows respect for patient individuality while maintaining the benefits of the Western scientific healing system [14]. By turning our perspective back to A.T. Still’s vision of osteopathy informed by Native American traditional healers and modern scientific concepts [1], this revolutionary reconsideration of historical OPP is another opportunity to demonstrate to mainstream Western medicine the richness and variety of osteopathic care and its associated scopes of practice.

Therefore, the purpose of the current commentary is to characterize the current challenges of an evaluation of historical OPP solely through the dominant Western biomedical lens and to introduce an anthropological perspective of the non-Western traditional/CAM legacy of OPP. Refocusing on patient values and expectations from a medical anthropology perspective allows for the consideration of diversity in sociocultural medical belief systems, thus fostering culturally sensitive, patient-centered osteopathic care. A framework for practitioners is also included, delineating what patients consider relevant for maintaining their health and well-being as part of evidence-informed osteopathic care in the Western secular clinical scenario.

## 2. Methods

The reporting framework used in the current essay followed established guidelines for writing a commentary [15]. Further, the essay was intended to help Western healthcare providers conceptualize and evaluate the relevance of philosophical concepts, historical principles, and non-Western sociocultural health assumptions common to several traditional/CAM perspectives so that they could improve their practice for the provision of ethical, and informed, PCC [16]. To address this overall goal, we considered the following two key questions from a medical anthropology perspective that were specific to the osteopathic profession:Are historical osteopathic principles and Western biomedical evidence both integrable into contemporary evidence-formed, culturally sensitive PCC?Is there an available framework to guide osteopathic practitioners to provide such inclusive approaches?

The theoretical framework for the current commentary was developed by a working group of experts [17] with at least 10,000 h of professional experience in education and scientific research (R.Z.-P., F.B., C.L., and D.D.) and in clinical osteopathic practice (R.Z.-P., F.B., and C.L.). More specifically, the framework was the result of a brainstorming process based on clinical observation and the best available evidence. To evaluate the rigor of the implemented methodology, a scale was used for a quality assessment of narrative review articles [18].

To identify eligible articles that would inform the current commentary, a literature search was performed between June and July 2022 in the following databases: MEDLINE (PubMed), EMBASE, and Google Scholar. The search terms (i.e., keywords: culturally sensitive care; osteopathic principles; manipulation, osteopathic; anthropology, medical; patient-centered care; medicine, traditional; complementary therapies; Western healthcare; health belief model) were adapted for each database, and suitable subheadings were used for each database searched. The search was limited to papers published in English. No limits were applied to the study design, population, study outcome, or date of publication. Reference lists from the articles were also searched, and a snowball procedure was used to identify the more relevant articles. To avoid placing any restrictions on the review and to capture the entire range of information about the topic, validity and quality assessments were not performed. When assessing the eligible articles, the authors (R.Z.-P., F.B., and C.L.) independently followed a two-stage selection process. First, each author independently read the abstracts of the eligible articles and then decided to include or exclude the article based on its relevance to the different elements of the commentary. Second, the full-text versions of the included studies were then processed using the same screening procedure used for the abstracts.

## 3. Results

The selected findings of the literature search were reported and grouped by pertinence into the five themes described below.

### 3.1. Historical Osteopathic Principles and Practices in Contemporary Care: A Legacy from Traditional/Complementary and Alternative Medicine

Traditional medicine is defined by the World Health Organization (WHO) as ‘the sum total of the knowledge, skill, and practices based on the theories, beliefs, and experiences indigenous to different cultures, whether explicable or not, used in the maintenance of health as well as in the prevention, diagnosis, improvement or treatment of physical and mental illness’ [19]. Western biomedical healing principles rely on a linear model based on cause and effect while traditional/CAM healing principles focus on returning each individual to health instead of targeting symptoms or causes [20]. Some indigenous traditional/CAM healing traditions have developed the Medicine Wheel in health and healing as a metaphor to represent four elements that must be targeted during treatment. [20]. The four quadrants of this culturally grounded conceptual framework represent the context (i.e., family, community, culture, and environment); mind (i.e., cognition, emotion, and identity); body (i.e., physical needs, and practical needs—including financial needs); and spirit (i.e., spiritual practices and teachings, dreams, and stories) [20].

Native American spirituality and healing practices may have influenced A.T. Still during his development of osteopathic medicine as suggested by recently released materials from the Museum of Osteopathic Medicine in Kirksville, Missouri (USA), that formally document his connections with the Shawnees, a former Northeastern Native American community; these experiences were also recounted in his autobiography [21]. Because a comprehensive description of the cultural specificities of the Shawnees, who had already been relocated from the Northeast at the time A.T. Still and his family lived among them at the Wakarusa Mission in Kansas, was published by Howard [22], information exists about their specific spirituality and healing practices. Further, although the indigenous groups across North America had their own ceremonies and rituals to treat their people, common principles existed across tribes to achieve healing through therapeutic approaches involving interactions of an individual’s body, mind, emotions, and spirit [23]. Interestingly, this concept of wholeness goes beyond each individual and extends to the interrelation of all living things (i.e., people, nature, and spirits) and is also represented in the sacred Medicine Wheel among the Plains Indians [20]. To restore health, the primary focus and concerns for treatment are placed on the ‘immortal soul’, which is symbolically placed at the center of the Medicine Wheel to foster balance among the body–mind–spirit–emotions components of the quadrants [20]. Another perspective Native American medicine healers still hold in common is that all diseases begin and end in the spirit of the person [24]. Such important sociocultural views from Native American spirituality and healing principles were used by Still when defining the osteopathic profession in the context of existing Western medical practices [25]. He described patients’ self-healing capacities and their body–mind–spirit connection, outlined the importance of the spiritual component for treatments that encompassed the belief in the immortality of the soul, and focused on restoring health versus targeting pathology and on multifactorial versus reductionistic etiology.

In 2002, Rogers et al. [26] published the most recent update of osteopathic principles and expanded the dualistic division of the body and mind. They also attempted to define what constitutes a person and included ‘spirit’ in that definition [27]. Thus, a body–mind–spirit approach uses art and science in the pursuit of optimal health rather than the absence of sickness [28], and it is characterized by a philosophical commitment to whole-person care that embraces the entire individual. In this approach, each person is considered an integration of physical, psychological, intellectual, and spiritual aspects that are equally important for health [28]. Therefore, with its traditional/CAM legacy and current evidence-based approach, the osteopathic profession is in a unique position to promote a scientific model of holistic care that is more inclusive of the rich variety of interpretations of historical OPP. Unfortunately, an anthropological interpretation to contextualize this information in patient care is missing.

Despite being regulated as a medical profession in the United States and as an allied health profession in many other countries, the WHO’s international publication on osteopathic training, which is a key document for the profession, was published in 2010 by its traditional/CAM department [29]. Two key historical OPP introduced in early Still publications were inherited from traditional/CAM and are still included in the WHO publication; specifically, they involve considering the patient as a dynamic interaction of the body, mind, and spirit and the human ability for self-healing [29]. However, the extent to which these historical OPP should be incorporated into Western healthcare for the provision of ethical and informed PCC remains unclear and is at the core of professional dissension.

### 3.2. Dilemma for Osteopathic Practitioners: Sticking with Historical Osteopathic Principles and Practices despite the Biomedical Evidence against Them or Focusing Only on Currently Available Evidence

Like allopathic physicians, osteopathic physicians in the United States are fully licensed and practice the full scope of medicine, including hands-on approaches guided by the concept of somatic dysfunction for diagnosis and treatment and for coding and billing purposes. Outside the United States, osteopaths are first-contact practitioners with limited rights of practice and are mostly restricted to hands-on approaches. Thus, defining a unified scope of osteopathic practice seems challenging since it differs based on specific country regulations in the existing healthcare systems [29]. For example, outside the United States, osteopaths can practice as allied health professionals and, therefore, are permitted to advertise for the treatment of conditions only when evidence of manual treatment efficacy is available, which means they mostly treat MSK-related conditions [30]. In such environments, such as the United Kingdom, osteopaths are required to refer patients when there is an insufficient or conflicting evidence base for manual effectiveness for non–MSK-related conditions [30]. In France, osteopathic practice is legally regulated and osteopaths hold a protected professional title shared by medical and nonmedical healthcare professionals and by nonhealthcare professionals [31]. Further, their scope of practice is legally defined as the treatment of patients based on their manual palpatory findings associated with somatic dysfunctions, i.e., a restricted MSK-related scope of practice [31]. In addition to these different types of regulations within the existing healthcare environments, there are ethical concerns in countries where osteopathic professionals are not regulated. For example, such practitioners can refer to historical OPP and claim they are treating MSK and non–MSK-related conditions without the obligation of providing evidence to support this wider scope of practice. Therefore, although included in international documents [29], the clinical relevance of historical OPP [26] remains unclear for osteopathy outside the United States.

Evidence-based models for MSK evaluation, diagnosis, and treatment are currently shaped by neuroscience and pain science [32,33]; therefore, different regulated practitioners may use similar therapeutic strategies based on the same biomedical evidence. In the manual therapy field, the current UK national guidance for noninvasive treatments for low back pain and sciatica states that osteopaths, chiropractors, or physiotherapists can apply only the recommended set of manipulations/mobilizations [34]. Consequently, ‘evidence-minded’ osteopathic practitioners will likely have the same scope of practice, propose therapeutic strategies for patients similar to other professionals, and refrain from using historical OPP not supported by evidence [7]. Further, this ‘evidence-minded’ group would likely endorse new frameworks for osteopathic care, such as the (en)active inference [35], that facilitate the integration of professional practices in Western care. This trend of using common evidence-based practices is currently being promoted by different manual professions towards an already existing set of professional practices. This professional specialty, orthopedic manual physical therapy (OMPT), is defined as ‘a specialized area of physiotherapy/physical therapy for treatment approaches including manual techniques and therapeutic exercises’ [36]. Similar to other professions sharing a similar biomedical epistemological framework, OMPT is also driven by the available scientific and clinical evidence and the biopsychosocial framework of each individual patient [36] and, as such, has no space for traditional/CAM healing practices. The removal of historical OPP from education and clinical practice seems to be the next logical step for professionals who want to focus on evidence-based care [7] and may be the most reasonable option in countries where osteopathy is regulated as a healthcare profession. It is also a more comfortable option for ‘evidence-minded’ practitioners who are openly marketing osteopathy as a profession that focuses on MSK care based solely on a Western biomedical healing system. In such environments, the next step is the merging of several professions towards an OMPT-like profession. However, some have expressed extreme viewpoints that belittle historical OPP when viewed from the Western biomedical dominant perspective and without an anthropological perspective [8]. Such professional attitudes may present ethical concerns for patients since the promotion of PCC based solely on practitioners’ Western values assumes their patients have the same values and likely fails to address the specific needs of individual patients. Failing to consider diversity in relation to patients’ underlying sociocultural assumptions of health displays a blatant lack of cultural sensitivity in clinical care and is the definition of ethnocentrism. Therefore, the currently missing anthropological perspective of historical OPP is important to address in the osteopathic profession to raise awareness of culturally sensitive care and to build an ethical and inclusive clinical framework for patients with Western biomedical and traditional/CAM belief systems.

Because modern healthcare involves multiethnic and racially diverse patient populations, a better understanding of cultural differences in medical beliefs and practices is necessary [11]. As such, medical anthropology, a subfield of anthropology, is relevant for clinicians promoting PCC. More specifically, medical anthropology is defined as the study of illness and health and the methods of healing in the context of cultural settings [10]. Medical anthropology also addresses the concepts of disease and illness to highlight the different perspectives of ill-health held by practitioners and patients [37]. On the one hand, disease, literally meaning dis-ease, refers to the patient’s biology and involves the perspective of the practitioner, who is trained to identify, label, and manage conditions. On the other hand, illness refers to the patient’s experience, i.e., how disease impacts their functioning, relationships, and social interactions shaped by the sociocultural environment [38]. The missing cross-cultural and comparative study of human behavior and culture and its influence on healthcare is apparent from the current evaluations of historical OPP, which were viewed through the ‘disease’ lens (i.e., an evaluation of the biological components) and failed to demonstrate clinically meaningful data and academically acceptable models. To address this failing, Esteves et al. [39], supported by various international groups, initiated a critical call for updates of the historical OPP theoretical frameworks for research, education, and evidence-informed practice. Evaluation of osteopathic models through the ‘illness’ lens (i.e., an evaluation from patient experiences) is a common feature of traditional/CAM and may provide more meaningful (qualitative) data. One pioneer of this field, Tyreman, investigated the anthropological–ecological narrative of osteopathy and suggested other ways of describing osteopathic care within the dominant Western biomedical environment [40].

To illustrate the relevance of exploring historical OPP through the ‘disease lens’ and also though the ‘illness lens’ where a culturally sensitive approach is crucial, we would like to posit an example of medically unexplained symptoms (MUS). Commonly observed in osteopathic care, MUS are persistent physical symptoms for which no conclusive organic explanation can be found. They are present in about 40–50% of all primary care consultations and about 50% of all secondary care consultations [41]. These MUS can cause mild to severe limitations in a patient’s daily functioning, as evidenced by lowered health-related quality of life scores and work problems [41]. However, the symptoms can be successfully managed when practitioners take a collaborative and inclusive approach to care by helping patients recognize the multitude of factors that may be affecting their lives and working with them to restore healthy functioning. According to Graver [42], osteopathic practitioners are in an ideal position to provide this type of comprehensive biopsychosocial approach to care for optimal outcomes in patients with MUS.

Paradoxically, the social and economic/political contexts associated with this clinical condition have hardly been researched, but Hanssen et al. [41] suggested the development of programs to promote a greater awareness of these contextual factors and to determine how their implementation could lead to better interventions for MUS. Traditional/CAM knowledge of holistic health may be a good option for care because it integrates the experiential praxis of a patient’s spiritual and physiological self with the relational praxis of the patient’s biological–sociocultural relationships during the conceptualization and delivery of health outcomes. Thus, integrative approaches combining Western biomedical and traditional/CAM perspectives, as presented in historical OPP, may result in the most appropriate care for groups of patients. Agarwal [43] recommended using traditional/CAM knowledge to inform the epistemological foundations of Western medicine by credentialing traditional/CAM practitioner teaching in allopathic healthcare institutions, providing faculty development at the existing allopathic health professional schools, and incorporating traditional/CAM content in allopathic medical education and practice. Scientific models have also been proposed that incorporate traditional/CAM practices and spiritual components of patients into care, thus establishing culturally sensitive, patient-centered, and evidence-informed care [44].

The chiropractic profession has similar issues with opposing professional viewpoints about the role and importance of historical principles and modern evidence-based practice in their profession [6]. In environments where practitioners have adopted modern evidence-based principles, external stakeholders have determined that such professional practice accords with modern healthcare principles and that it should be used by legitimate healthcare practitioners for better integration into Western healthcare systems. In contrast, practitioners who are reluctant to use the evidence approach and are guided mostly by historical principles are unlikely to make practice changes in the absence of evidence that substantiates their claims [6]. The authors likened this situation to that of an unhappy couple that stays together for reasons unconnected with love or even mutual respect, despite differing worldviews and the option of an amicable divorce to resolve the issue [6].

Continuing the couple analogy from the chiropractic profession, a possible option for the osteopathic profession would be to use something like integrative behavioral couple therapy to address conflicting viewpoints before making any irreversible decisions. The main therapeutic strategies would involve empathic joining, an expression of soft emotions, acceptance, perspective change, and psychological distancing [45]. Instead of focusing on the existing problems, which usually reinforces them, we should try to reestablish common bonds, i.e., providing the best available care for patients, that would be beneficial for the whole profession. Therefore, the inclusion of cultural sensitivity would better inform how each group interprets ‘patient-centeredness’ and is worth investigating. For example, to focus on the diversity of patient values and expectations, we would first need to determine whether they belonged to the Western biomedical or traditional/CAM belief systems. Then, we could discuss how to include them ethically within a Western evidence-informed clinical scenario.

### 3.3. Moving beyond the Current Polarity between Osteopathic Practitioners: Refocusing on Patient Values and Expectations from a Medical Anthropology Perspective

The ‘illness’ perspective involves the patients’ experiences that, contrary to the biological component of the ‘disease’ perspective, can be shaped by their sociocultural environment and are, therefore, prone to the diversity of patients’ underlying health belief systems. The WHO defines ‘health as a state of complete physical, mental, and social well-being and not merely the absence of disease or infirmity’ [46]. Similar to a traditional/CAM perspective, wellness targets the interconnections of each individual’s emotional, mental, physical, and spiritual health [47]. According to Hey et al., emotional health refers to the sum of emotional states at any given time; mental health refers to the ability to act on information, clarify values and beliefs, and exercise the decision-making capacity; physical health refers to the ability to maintain an awareness and knowledge of nutrition and exercise, to monitor symptoms, and to understand recuperative capacity and the prevention of injuries; and spiritual health involves the need for meaning, purpose and fulfillment, and inner strength [47]. With their emphasis on proper MSK system functioning to promote health and resist disease processes [29], historical OPP [26] promote a holistic approach to health that is at the anthropological core of traditional/CAM [48], i.e., vis medicatrix naturae, and they characterize healthcare providers as the facilitators of these natural healing processes [49]. In support of this approach that enables the incorporation of all patients’ sociocultural health assumptions, Tyreman [40] introduced the anthropological–ecological narrative where two key aspects of the osteopathic profession were explicitly recognized: considering each individual as an organism rather than a sum of mechanisms and placing the clinical focus for healthcare on a person rather than on a disease. As such, this new perspective provides a foundation to understand and incorporate the diversity of patients’ and practitioners’ beliefs into osteopathic care in relation to health, disease, function, and dysfunction. Understanding how different factors influence the representation and organization of the sociocultural environment and the associated socially learned behaviors remains a central concern of anthropology. Therefore, investigating the diversity of patients’ beliefs and expectations may be crucial when seeking osteopathic care since historical OPP navigate between Western biomedical and traditional/CAM sociocultural health assumptions [50].

Medical anthropology addresses the relationship between health and the individual, considers the narrated experiences of illness and how suffering takes place within cultural and social institutions [10] and, as such, offers a path to resolve the polarity within the profession. Further, this discipline may offer insightful perspectives on how traditional/CAM have developed in Western societies. Dissatisfaction with the excessive emphasis on ‘dis-ease’ by the Western biomedical system was a primary reason for patients seeking care from traditional/CAM practitioners, who typically pay greater attention to the patient’s experience of being ill and appreciate the role of social factors [38]. When considered from a medical anthropology perspective, the answers to the following six questions typically determine patients’ and relatives’ experiences of illness and the associated behaviors: (1) what has happened? (2) why has it happened? (3) why me? (4) why now? (5) what would happen if nothing was done about it? and (6) what should I do about it or who should I consult for additional help? [37]. This kind of focus on patients’ experiences and values allows them to be considered as an individual and has become one of the four dimensions of PCC in Western care [13]. Its use in clinical practice reduces healthcare costs and improves patient outcomes [51]. Further, PCC incorporates the patient’s perspective as part of the therapeutic process and highlights the need to communicate in a manner that creates adequate conversational space to elicit the patient’s agenda (i.e., understanding the impact of pain, and their concerns, needs, and goals), which ultimately guides clinical interactions [13]. As such, PCC has been endorsed by ‘traditional-minded’ and ‘evidence-minded’ osteopathic practitioners, but different interpretations are arising from diverse patients’ understanding of their health and wellness that, based on their sociocultural beliefs, may not be limited to alleviating MSK symptoms.

Recently, Shaw et al. [52] reconceptualized the therapeutic alliance in osteopathic care. They argued that physical bodies are imprinted with biographical and cultural meaning through learning processes that start before humans learn language [52]. They highlighted the importance of co-constructed narratives that depend on the practitioner’s ability to find points of entry into a patient’s world and that require narrative competence in imaginative thinking and radical listening skills [52]. Although they pointed out that sociocultural dynamics influence individual beliefs and behavior and cooperative communication (i.e., talk, touch, and body language), they did not discuss the diversity of sociocultural factors and their importance in shaping narratives that make sense to patients, especially for those with traditional/CAM belief systems [52]. From a multicultural perspective, it is crucial that practitioners consider different patient values and expectations because they directly influence the therapeutic outcome [9].

Similarly, most osteopathic research focuses on a Western model of care. For example, a large randomized controlled trial evaluated the specific effects of standard osteopathic manipulative treatment (OMT) versus sham OMT in patients with chronic low back pain [53]. The authors found that standard OMT had a significant effect on activity limitations specific to low back pain, but they were not perceived by patients [53]. Although the specific effects of OMT can be measured, their clinical relevance is questionable. Therefore, researchers should redirect their attention toward nonspecific effects associated with OMT, and clinicians should redirect their practice toward person-centered rather than body-centered (i.e., finding and treating somatic dysfunctions) osteopathic care [54]. This current trend in research and practice highlights the importance of the nonmanual components of osteopathic care versus the historical manual components (e.g., OMT) that are the hallmark of the profession. Because OMTs are triggers of placebo or nocebo effects, the role and importance of contextual factors have been investigated for general Western healthcare [55], for Western healthcare specific to MSK treatment [56], and in traditional/CAM care [57].

However, the biomedical approach continues to struggle with understanding the complex processes underlying placebo responses and the potential interactions between the placebo mechanisms and the pharmacological effects of treatments. To advance the medical anthropology perspective, a distinction needs to be made between placebo and nocebo effects in clinical practice and placebo and nocebo responses as outcomes in clinical research trials. Placebo and nocebo effects refer to the beneficial or adverse effects that occur after the administration of an inert treatment or as part of active treatments, and they are typically attributable to such mechanisms as patient expectations [58]. A placebo is more than an inert pill or intervention that produces no ’real’ effect. A better way to conceptualize placebo effects would be to view them as a process that produces somatic effects (including healing responses) through any other factor than the intended or postulated mechanism of action for the intervention. These effects represent complex and distinct psychoneurobiological phenomena from behavioral, neurophysiological, perceptive, and cognitive changes that occur during the therapeutic encounter [59]. In Western medicine, the placebo effect is considered a nonspecific process that needs to be controlled, but in traditional/CAM, it is considered a specific effect of a healing ritual [60]. Although some might suggest a performative characteristic in healing rituals, Western medicine could also be considered as a sociocultural healing ritual [60]. In general, healing rituals are considered a sense-making process for patients because they heal and restore their world through a symbolic re-editing of their body and self-image [61].

Medical anthropologists have identified the use of narratives as one of the primary processes for recreating meaningful order from the disorder of illness [61]. For example, Gukasyan and Nayak [62] summarized four common contextual factors shared by various healing traditions: (1) the therapeutic relationship; (2) the healing setting; (3) the rationale, conceptual scheme, or myth; and (4) the ritual enactment. Therefore, practitioners should develop essential skills to perform healing rituals that combine body experience symbols with a narrative made of verbal symbols [61]. Exploring these common responses from an anthropological perspective, i.e., how similar physiological processes are interpreted differently according to patients’ sociocultural health assumptions, may increase understanding of the relevance of historical OPP in relation to the traditional/CAM legacy and contemporary care in the dominant Western healthcare environment. Anthropologists believe placebos and nocebos are culture-bound because they do not exist in a vacuum. Therefore, the occurrence and magnitude of placebo or nocebo effects will depend on the wider context of cultural beliefs, values, expectations, assumptions, and norms; they also depend on the social and economic realities in which they occur [57]. Contextual factors, which include patient–practitioner interactions, the social observation of others, and environmental cues, can also influence placebo or nocebo effects and may be implemented to enhance treatments [55]. Universal mechanisms may also contribute to more effective interventions and therapeutic encounters. Examples of such mechanisms include ritual and technical practices, shared symbolic and mythological elaboration, the charisma of the healer, social legitimization and validation, and the rewarding feeling of having earned healing by enduring a difficult experience. Further, suggestibility may have a central role in the therapeutic efficacy of these mechanisms. Unlike discrete psychological traits (e.g., hypnotic suggestibility), suggestibility is characterized by decreased effort control and increased susceptibility to contextual factors and often involves framing effects, verbal and nonverbal suggestions, and peer influences. In manual care, contextual factors are actively interpreted by the patient and may elicit expectations, memories, or emotions that in turn influence the health-related outcomes, such as placebo or nocebo effects [56]. Recently, an international consensus group investigated the implications of placebo and nocebo research in healthcare practice and proposed the first step toward the development of evidence-based and ethical recommendations [58]. The words used by osteopathic practitioners can have a powerful effect on patients, and the problematic absence of professional standards regarding the narratives associated with historical OPP needs to be addressed respectfully and ethically to avoid nocebo effects and improve patient care.

Because healthcare has been built through rituals and symbols, it has the ability to induce suggestibility among patients. A person’s propensity to respond positively to suggestions, i.e., thinking and acting on the suggestions of others, is a key feature of traditional/CAM procedures [63]. A primary role of traditional healers is enhancing the patient’s susceptibility to the influence of external interpretations of personal experiences, contributing to the dynamic transmission of health beliefs that are later validated by patients [63]. These authoritative therapeutic postures, which are typical of traditional/CAM and early osteopathic practitioners, can be understood within a specific sociocultural environment; however, modern Western care fosters a more collaborative approach through shared decision-making processes. Paradoxically, authoritative practitioners of historical OPP in contemporary care appear to promote integrative approaches, while evidence-informed practitioners now engage in more collaborative approaches. Therefore, delineating an ethical framework to describe the acceptable therapeutic attitudes of ‘traditional-minded’ and ‘evidence-minded’ osteopathic practitioners should be a priority for the profession to conscientiously and ethically manage contextual factors for the patient’s benefit that enhance the placebo effect and avoid nocebo effects. As outlined by Rossetini et al. [56], both effects can happen during any clinical phase (e.g., consultation, examination, and treatment) and affect the symptom perception, experience, and meaning.

With increased diversity in healthcare, first-contact practitioners, including osteopaths, are expected to provide sensitive and culturally appropriate care to patients and families from distinct cultural and social backgrounds [64]. People from different cultures have their own beliefs and ethical, social, and moral values that must be respected and valued, and although providing effective and culturally sensitive care may be daunting, practitioners need to be aware of the personal, cultural, and social beliefs and preferences of their patients so they can be incorporated into care as defined by both PCC and evidence-based practice.

### 3.4. Historical Osteopathic Principles and Practices: Avoiding Western Biomedical Ethnocentrism and Promoting Culturally Sensitive, Person-Centered Care

Practitioners consider patient beliefs, expectations, and prior experiences of treatment when planning person-centered care because these factors can influence the therapeutic outcome [56]. In the Western healthcare environment, the diversity of individual sociocultural beliefs, perceptions, and values related to health and wellness have shaped specific social and therapeutic frameworks [47]. Therefore, patients’ sociocultural health assumptions will affect the value of manual therapies differently when striving for positive (physical) health and wellness outcomes. Additionally, an effective therapeutic alliance using appropriate communication about expectations is an indicator of improved patient outcomes [52] and can be strengthened by examining psychological, social, and lifestyle issues; communicating health information; and coaching patients to modify behavior in ways that make sense to them [65]. A review by Asnaani and Hofmann [66] investigated the empirical findings and common features that enhance therapeutic collaboration in a multicultural setting, and the authors suggested guidelines for achieving this goal with patients and practitioners from various cultural and racial backgrounds. To highlight the complex nature of multicultural care in a Western environment, Hays proposed the acronym, ADDRESSING (Age and generational influences, Developmental disabilities and Disabilities obtained in later life, Religion and spiritual orientation, Ethnic and racial identity, Socioeconomic status, Sexual orientation, Indigenous heritage, National origin, and Gender), to serve as a reminder to practitioners about the multifaceted nature of multicultural therapy [66]. This acronym represents the diversity of cultural identity and the cross-cultural factors practitioners should consider in culturally sensitive care [66]. Although the implementation of this framework in osteopathic clinical practice may benefit patients, the only available evidence is for empirical findings from investigations of the benefit of specific components of culturally sensitive care, which predict maximal clinical benefit for patients [66].

Because racial inequities in health and healthcare are well documented, a recent study investigated the knowledge, beliefs, and experiences of osteopathic students in relation to race-based medicine [67]. Importantly, the authors clarified that race is not a biological or genetic concept, but a social construct to categorize people based on observable traits, behaviors, and geographic location [67]. Before the study, only half of the students had heard of the term race-based medicine and only 44.4% provided the correct definition for it [67]. According to the authors, these findings are important because familiarity with race-based medicine may be an indicator of structural racism in healthcare education, which may contribute to negative patient outcomes (e.g., emotional distress, and feelings of isolation) [67]. However, evidence suggests that increased cultural sensitivity, an ability to effectively communicate cross-culturally with patients, and a greater knowledge of diverse cultures are critical for improving the care of patients with diverse cultures and ethnicities. Transcultural care enables practitioners to value the social and cultural needs and preferences of patients and their families, which is essential to reduce healthcare disparities caused by the disparate social and cultural values and beliefs of patients, practitioners, and healthcare practices in general [64].

From an osteopathic professional perspective, transcultural care can be applied to patients and practitioners with different dominant sociocultural health assumptions, i.e., Western biomedical or traditional/CAM. Because the sociocultural health assumptions of patients are not fixed or assigned to specific racial or ethnic groups, it is important to stress that they can evolve over time. A good example of this evolution is the development of so-called shamanic tourism in the last twenty years, where many Westerners have traveled to distant countries to participate in exotic therapeutic practices that are a form of traditional/CAM. Labeled as shamanic tourism [68,69,70] or ethnomedical tourism [71], these temporary migrations have resulted in numerous studies that have included them in the framework of the evolution of tourism practices. It is likely that these Western ‘shamanic tourists’ are patients who have traditional/CAM perspectives for maintaining and improving health and well-being and are interested in care that Western practitioners do not provide. From a traditional perspective, the historical OPP in use today continue to represent the seminal integrative approach introduced by A.T. Still in the dominant Western environment, which combined Western biomedical and traditional/CAM healing wisdoms. This integrative approach is well represented in traditional/CAM cultures by the Andean prophecy of the Eagle and the Condor [14]. The condor symbolizes the people from the South, who were led by spirit-based healing wisdom, and the eagle symbolizes the people from the North, who were led by scientific-based healing wisdom; when the two fly together again, they will bring a more balanced healthcare perspective throughout the world [14] (Figure 1).

This trend may be similar to patients who sought out early osteopathic practitioners because their care was different from the dominant Western biomedical paradigm that did not meet patient expectations. A key characteristic for determining a patient’s dominant sociocultural health assumptions is their willingness to seek care without physical symptoms. Historically, these non-symptomatic scenarios were handled by osteopathic practitioners, but ‘evidence-minded’ practitioners are currently challenging them. Further, these clinical scenarios have become an ethical concern because a clear framework is still lacking, and a nocebo narrative could be easily introduced by practitioners [55]. However, prevention is a key element from the traditional/CAM perspective, and a salutogenic approach is usually preferred to maintain health instead of seeking care only when necessary [72]. Although evidence-based practice incorporates patient values and expectations into care, controversy exists about the inclusion of health belief systems from non-Western patients, and this issue has raised ethical concerns, particularly for ‘traditional-minded’ practitioners who should become fully aware of the implications of using historical OPP inherited from traditional/CAM principles.

Similar ethical issues have been investigated through the content analysis of the discourse of traditional/CAM practitioners in clinical care [43]. Agarwal [43] observed that traditional/CAM practitioners use the epistemological foundations of legitimization in conjunction with identity, sense and intuition, and environment and community to situate the meanings of holistic health within the normative discourse. Instead of proposing a radical opposition to the Western dominant healthcare paradigm, epistemologies defined holistic health by organizing diverse knowledge foundations through the reconciliation and integration of differences and included diverse modes of evidence, such as nonempirical forms of whole-body experiences, privileging the relational praxis through an integration of the individual’s biological and sociocultural environment. Traditional/CAM knowledge of holistic health is able to integrate the patient’s spiritual experience and personal biological, sociocultural, and epigenetic relationships within the conceptualization and delivery of healthcare [43]. A similar approach would be beneficial for osteopathic practitioners so that patients values and expectations are met with a narrative related to historical OPP rather than one focusing on pain control through person-centered, OMPT-like Western biomedical approaches. Importantly, the inclusiveness of such a framework, as part of a formal, culturally sensitive approach to PCC, will not mean the rejection or promotion of one type of practice. It will instead allow ‘traditional-minded’ and ‘evidence-minded’ practitioners to focus on patient values and expectations within an evidence-based approach. As such, Agarwal [43] proposed that conceptualization of these integrative models of care may result in a greater sensitivity and awareness of the practice of healthcare and the maintenance of health. Nevertheless, ‘traditional-minded’ practitioners should fully understand the duties of being a regulated healthcare professional and their obligation to work within a clear and ethical framework. For example, osteopaths in the United Kingdom are required to be aware of the professional duty of candor, i.e., to be open and honest with patients, colleagues, relevant organizations, and regulators [73]. Specifically, the Osteopathic Practice Standard A1.4 states, ‘Be aware that patients will also have particular needs or values in relation to gender, ethnicity, culture, religion, belief, sexual orientation, lifestyle, age, social status, language, physical and mental health and disability. You must be able to respond respectfully and appropriately to these needs’ [73]. This standard highlights the importance of being able to provide the culturally sensitive, patient-centered osteopathic care that has been suggested throughout the current commentary.

By focusing on a person rather than a disease, historical OPP introduced the concept of PCC that is now common in Western medicine, and this framework has been progressively incorporated into other medical fields [39]; however, the widespread use of patient-centered approaches in various healthcare professions currently challenges this defining feature of historical OPP from the Western biomedical dominant perspective [74]. The ability to implement culturally sensitive, patient-centered osteopathic care that incorporates patients’ traditional/CAM health belief systems in a Western healthcare setting appears to be an important way to promote the clinical relevance of historical OPP in contemporary care. This supposition should be considered as a form of transcultural care. For example, Shahzad et al. [64] developed a comprehensive understanding of practitioners’ challenges and approaches to the provision of transcultural care for patients with diverse ethnicities. They identified four challenges to transcultural care: (1) alleviating the intrapersonal struggles of practitioners who want to provide effective care but are doubtful, (2) addressing cultural conflicts that diverge with patients’ and practitioners’ views and expectations, (3) understanding varied expressions of suffering and combating uncertainties, and (4) navigating personal and organizational constraints [64]. The authors also identified three approaches for providing effective transcultural care: (1) practicing self-criticism and tolerating differences, (2) enhancing interpersonal and psychological skills, and (3) collaborating with peers and family caregivers [64]. Further, skills identified for nurses could be easily adopted by the osteopathic profession to help ‘traditional-minded’ and ‘evidence-minded’ osteopathic practitioners better understand that they are probably offering care to patients who want different types of treatment because they have different underlying sociocultural health assumptions. A set of guidelines may help both groups understand their professional values and reduce the Western biomedical ethnocentrism interpretation of historical OPP, to refocus on patient values and expectations, and to offer culturally sensitive patient-centered osteopathic care (Table 1).

Other Western healthcare professions have applied a meta-framework capable of considering multiple standpoints to solve the incompatibilities between professional viewpoints [75]. Thus, to address multiple theoretical and philosophical perspectives within a discipline, psychologists and nurses introduced a practical perspective called integral theory [75]. In practice, integral theory makes space for the patient’s emotional, spiritual, and mental needs by incorporating self-preservation, self-adaptation, self-immanence, and self-transcendence into care. Liem and Lunghi [75] proposed this model as a way to distance care from the subject–object relationship, expressed through the patient–practitioner duality, and to implement shared decision-making through a participatory treatment process that helps clinicians consider all areas of human experience in osteopathic care. However, the authors did not consider the impact of different practitioners’ perspectives, i.e., ‘traditional-minded’ versus ‘evidence-minded’, on clinical care, and instead focused on overcoming the artificial fragmentation of dysfunctions into their corresponding somatic, psychological, cultural, and energetic aspects [75].

Considering the diversity of patients’ sociocultural health belief systems, meeting the specific expectations of patients may be challenging for osteopathic practitioners [29,76]. Therefore, patient expectations of osteopathic care should be investigated, ranging from the treatment and prevention of MSK-related conditions [77] within a Western biomedical framework to nonspecific support for well-being and health within a traditional/CAM framework [78].

### 3.5. The Cynefin Framework: An Inclusive Approach to Guide Culturally Sensitive, Evidence-Informed, Person-Centered Osteopathic Care

The purpose of this commentary is not to cause strife between Western-based and traditional/CAM knowledge but to balance the current strong and opposite opinions in the osteopathic profession regarding the available evidence and its use to inform PCC (Table 1). Without transcultural skills or even an awareness that patients may hold different sociocultural assumptions of health, some osteopathic practitioners will continue to value only Western biomedical evidence to inform their practice and increase the odds of falling into what has been described as ‘Western epistemological racism in healthcare’, i.e., conscious or unconscious attitudes nurturing the domination of Western-based knowledge over other knowledge [79]. Therefore, future exploration of the clinical relevance of historical OPP, rooted in traditional/CAM knowledge, should primarily investigate clinical outcomes with the PCC framework to incorporate the diversity of patients’ sociocultural beliefs systems towards recovering and maintaining their health and wellness [78]. This commentary is not intended to promote the use of historical OPP to embrace the existing movements in healthcare that decolonize and indigenize the Western biomedical paradigm [80]. Rather, historical OPP should be investigated as an integrative approach for patient care to delineate a larger scope of practice in the manual therapy field when compared with other OMPT-like professions that rely exclusively on Western evidence focusing on MSK care. Within the Western healthcare environment shaped by biomedical frameworks, historical OPP should be considered as powerful professional tools to structure and promote culturally sensitive, patient-centered, evidence-informed care, thus becoming strong pillars for the future of the osteopathic profession (Figure 2).

As part of this inclusive approach, the Cynefin framework is proposed to help osteopathic practitioners understand culturally sensitive, patient-centered care (Figure 2). The different sets of therapeutic actions and related narratives are presented in the five domains as options to help patients make sense of expected changes in their bodily perceptions according to their dominant underlying sociocultural health assumptions.

A variety of osteopathic care can be proposed to patients according to the beliefs, preferences, and expectations associated with their underlying sociocultural health assumptions. Further, patients may seek osteopathic care to promote health with or without physical symptoms [81]. Health has been described as a silent experience because patients have little awareness that they are healthy until symptoms require their attention [72]. Because the way of perceiving, expressing, and controlling pain is a culture-specific learned behavior, an anthropological exploration of pain may benefit practitioners by improving their understanding of human pain and suffering beyond their local setting and knowledge of the source and variety of meanings and consequences of suffering. A biopsychosocial–spiritual approach to MSK symptoms [82] was proposed and explicitly included affective, behavioral, and cognitive spiritual dimensions to optimize the therapeutic alliance [50]. The application of this approach may refine PCC and provide a clinical application of historical OPP in contemporary care; however, this approach must be consistent with the ethical principles of healthcare pluralism to support equitable knowledge representation [43].

Despite academic acceptance of the biopsychosocial approach, much of osteopathic practice remains firmly grounded in a biomedical model [83,84]. According to Naylor et al. [85], if the shift to person-centeredness constitutes a holistic approach beyond biomedical and biopsychosocial models, then the siloed focus of MSK outpatient practice on individual body regions may leave it lagging behind. One possible way to avoid this outcome and achieve a more desirable shift is through the adoption of a narrative approach that empowers patients. Indeed, narrative-based and person-centered practices have emerged in response to the perceived shortcomings of the Western biomedical approach [85]. The implementation of different skills, knowledge, and professional values in clinical care is complex but crucial for positive patient satisfaction and clinical outcomes. A recent study among osteopathic practitioners investigated decision-making processes when selecting different approaches for clinical care, and the content was based on a theoretical model of three therapeutic approaches—the treater, communicator, or educator—that shifted perspectives from specific manual care to the importance of the narrative to help patients make sense of their bodily perceptions during treatment [86]. The clinical rationale for exploring the role of different narratives associated with historical OPP, such as the body–mind–spirit interaction and the self-healing capabilities of each person, is to help patients make sense of their personal experience of physical symptoms. This approach would be especially beneficial for patients who have not perceived positive outcomes from previous care that relied solely on Western-based epistemological frameworks (Figure 2). The same perspective applies to different healing rituals commonly found in osteopathic care, such as evaluating and treating non-symptomatic body regions, to build specific therapeutic alliances that help patients make sense of their physical ailments.

When using PCC with patients, osteopathic care requires different decision-making processes and therapeutic roles for practitioners, and Tyreman [72] introduced a tool to manage this complexity. From a Welsh word for habitat, the Cynefin framework was proposed to assist decision-making processes in management and when complexity challenges insight, prediction, and decisions [87,88]. This framework was then introduced in the medical field [89,90] and later in osteopathic care by Lunghi and Baroni [91] to inform clinical reasoning and decision-making processes. When following the Cynefin framework, different osteopathic manipulative techniques can be proposed to patients as manual procedures that influence their bodily perceptions within a specific epistemological framework shaped by their primary sociocultural health assumptions (Figure 2). For example, passive manual approaches with a minimal use of direct, indirect, or combined techniques on the body or a maximal use of systemic, homeostatic-adaptogenic techniques on the impaired body system may be proposed and combined with active approaches, such as lifestyle counselling, exercise, and nutritional advice, and with top-down strategies, such as mindfulness for stress management [91]. Further, person-centered osteopathic care that includes the spiritual dimension in healthcare can be proposed as another top-down strategy when using this framework. Importantly, the four domains of the Cynefin framework may illustrate the current symptom or non-symptom-based approaches in osteopathic care. In the simple and complicated domains, a symptom-based model drives the decision-making process for evaluation, diagnosis, and treatment; and management strategies are usually selected from the evidence and practitioner experience of similar clinical contexts. In the complex and chaotic domains, additional considerations—narratives about individual meaning, purpose, and significance concerning the self, family, proximities, community, nature, and the sacred expressed through beliefs, values, traditions, and practices—can be incorporated in the healing ritual during the evaluation, diagnosis, and treatment. The multimodal integration of interoceptive, proprioceptive, and exteroceptive information and spatial-contextual features shapes the meaning and psychophysiological impact of touch [52]. This sense-making process can be embedded in different narratives, depending on the Western or traditional/CAM underlying the sociocultural health assumptions of patients. As part of the process of achieving knowledge of a complex adaptive system, the Cynefin framework also considers situations where there is no clarity about which of the four domains apply, i.e., the disordered or confused space that is graphically represented at the center of the framework between the simple/complicated and the complex/chaotic domains [92]. The confused space helps the perspective move from overidentifying with the practitioner’s mindset and knowledge, and it highlights the importance of the patient’s intuition for clarifying the integration of psychobiological functions with the existential and spiritual domains. Although a confused space seems to represent an irresolvable internal contradiction or logical disjunction, it can also be considered as a safe and contemplative space to help the patient reflect on difficult, undecided situations and connect with possible solutions. From there, patients could become more integrated into their specific context and environment while practitioners promote meaning and personal growth that allow their patients to develop values and an improved body awareness through manual and nonmanual therapeutic approaches [92].

Patients with chronic pain may be a good clinical example to illustrate this confused space. They are likely to have poor body awareness and be exhausted and disappointed by previous healthcare professionals, especially if they have different sociocultural health assumptions that were not previously considered [93]. As advocated by Louw et al. [93], clinical scenarios such as this require more care so that patients can make sense of their illness and the proposed therapeutic strategies. The best way to help these patients is to create a therapeutic space that promotes sense-making. When such clinical scenarios are in the confused space, the Cynefin framework may help practitioners to stop overidentifying with their knowledge and intuition and to start helping patients in the sense-making process, such as providing an emergent perspective from a shared decision [92]. Further, time is also necessary for a better understanding of patient intuition about the integration of different processes for the body, function, and existential domain, which can be perceived differently depending on patients’ underlying sociocultural health assumptions [92].

By using nonverbal behavior, proximity approaches, interoceptive touch, and mindful-based procedures that support effective communication, a better therapeutic alliance during osteopathic encounters can be created [94]. For example, touch-based strategies are valuable for creating collaborative agreement related to goals and tasks and for the development of successful relationships and cooperative communication, especially for patients confused by sociocultural health assumptions outside their usual worldview [78,91]. Touch also has a role in the development of synchrony through a more precise categorization of individuals, where more adaptive feedback loops are created to minimize surprise, increase understanding, and reduce physical and psychological stress, all of which are crucial for daily living [95,96,97].

As suggested by historical OPP, such as the body–mind–spirit osteopathic tenet, the inclusion of the spiritual dimension in healthcare in the confused space of the Cynefin framework may promote a better understanding and balance of patients’ and practitioners’ sociocultural health assumptions (Figure 2). To avoid the misinterpretation of patient values and expectations, practitioners have different options from the different domains to manage clinical complexity and foster a stronger therapeutic alliance through a person-centered approach. Because varying patient expectations and needs exist in osteopathic care, the right questions need to be asked during shared decision-making processes. From the patient’s perspective, the ability to engage meaningfully with a given therapeutic approach requires their belief in the treatment’s effectiveness for their specific situation or in the approach used to achieve a certain level of person-centeredness; however, there also needs to be a treatment ‘fit’ with the individual patient’s lifestyle [85]. This idea of paying careful attention to individual patient narratives diverges from Western biomedical roles of practitioners, which prioritize the diagnosis and management of physical impairments [85]. To acknowledge the patient as an individual, practitioners should verbally and nonverbally meet their patients as equals, validate their experiences, and individualize their treatment [51]. Further, practitioners need to be open, reflective, aware, and responsive to verbal and nonverbal cues, providing an even balance between engaging with the patient (e.g., eye gaze) and writing/typing notes during the interview [51].

According to Shaw et al. [52], osteopathic practitioners have the potential to positively influence patients’ health beliefs, body awareness, and previous experiences that influence avoidant behavior. Unfortunately, there is limited information about how positive changes occur or on the barriers that limit changes for patients and practitioners. The research suggests that specific communication, i.e., being verbal or nonverbal about osteopathic manipulative techniques, builds alliances through the social ‘ritual of the therapeutic act’ [52]. To better meet patient values and expectations, we need to accept that historical OPP are associated with a specific narrative and manual skills inherited from traditional/CAM and that they are different from MSK evidence-based professional skills. Thus, different contextual factors present during the therapeutic encounter between the patient and practitioner, such as healing rituals and signs, may trigger placebo or nocebo effects [52]. For that reason, the adoption of medical anthropology, a discipline that studies health and illness in the context of sociocultural settings, may help osteopathic practitioners better describe diverse patient expectations according to their underlying Western or traditional/CAM health assumptions (Figure 2).

Like responder analyses that investigated clinical predictors to target patients who were likely to improve after osteopathic care [98], the inclusion of new patient-reported measurements that evaluate patients’ underlying sociocultural health assumptions, ranging from Western-based to traditional/CAM perspectives, may outline narratives and healing rituals associated with positive clinical outcomes. Such qualitative data would build a specific epistemological framework for the use of historical OPP in the Western healthcare setting and identify the patient subgroups most likely to improve with its use. Similar to epistemologies illustrating how the traditional/CAM narrative and healing rituals are positioned as alternative within the sociocultural context of patients, ‘traditional-minded’ practitioners who rely on historical OPP should actively embrace the challenge of carving out a distinct knowledge space that reflects part of our professional identity in the current Western evidence-based environment.

## 4. Discussion

Historical OPP are being challenged by current Western biomedical evidence that focuses on disease as a biological component and from the practitioner’s perspective [39]. Thus, two different professional groups coexist within the osteopathic profession: ‘traditional-minded’ practitioners who follow historical OPP despite evidence against those models and ‘evidence-minded’ practitioners who are moving towards OMPT. Both routes to practice and patient care raise concerns about the long-term specific and sustainable professional identity within the dominant Western healthcare environment. To move beyond mainstream Western biomedical ethnocentrism in manual therapy, an anthropological perspective to evaluate patient values and expectations should be incorporated into osteopathic care. In particular, the therapeutic alliance (i.e., healing rituals and narratives) and manual procedures should be expanded to provide culturally sensitive PCC, which is the cornerstone of care for Western healthcare [65]. Refocusing on cultural awareness and sensitivity will enable us to reflect on implicit biases and recognize their potential negative effects on quality care; such reflections will also allow osteopathic practitioners to better meet diverse patient values and expectations [64]. In addition, a specific narrative associated with historical OPP could be included in some traditional/CAM epistemologies to help patients make sense of nondominant traditional/CAM sociocultural health assumptions. We believe this unique integrative approach between Western biomedical and traditional/CAM traditions was historically present at the inception of the osteopathic profession [1].

In the present commentary, the categorization of ‘traditional-minded’ and ‘evidence-minded’ osteopathic practitioners was used to illustrate an extreme professional posture. We included discussion of the missing anthropological perspective of historical OPP to help the profession move beyond these conflicting viewpoints and the inflexible silo thinking that is unhelpful for patients, practitioners, educators, researchers, and policy makers, particularly when describing the scope of practice of regulated healthcare professionals within a dominant Western environment.

To the best of our knowledge, the current essay is the first to specifically describe an anthropological perspective of historical OPP. Further, our medical anthropological perspective of the legacy of traditional/CAM principles in historical OPP is offered to advance the osteopathic profession by promoting ethical, culturally sensitive, evidence-informed PCC in a Western secular environment. Such inclusive approaches are likely to meet patients’ values and expectations, whether informed by Western or non-Western sociocultural beliefs, and to improve their satisfaction and clinical outcomes; however, we acknowledge some limitations associated with our approach to this topic. For example, the description used to represent the current challenges in the osteopathic profession was based on studies specifically chosen to promote discussions among practitioners and to raise awareness about the need to promote inclusiveness through culturally sensitive approaches; therefore, future research should be grounded in this anthropological perspective. Further, the profession should consider hosting a consensus conference that includes the broader involvement of international osteopathic groups, such as clinicians, educators, patients, stakeholders, and other healthcare professionals [99,100]. A first step of this consensus process would be to qualitatively and quantitatively describe the current challenges between ‘traditional-minded’ and ‘evidence-informed’ groups regarding the relevance of historical OPP in contemporary care.

## 5. Conclusions

Cross-cultural competencies have been introduced in Western healthcare education to help clinicians reinforce the therapeutic alliance and improve patient outcomes; however, raising awareness of the importance of such skills in clinical care may be challenging for practitioners who have not been exposed to patients with non-Western sociocultural health belief systems. To address these challenges and the differing viewpoints among healthcare professionals, the current commentary was intended to highlight the current practice of the osteopathic profession through the lens of Western and non-Western sociocultural health belief systems.

All osteopathic practitioners share the same interest in improving patient health and wellness, but the diversity of sociocultural health assumptions, from both practitioners’ and patients’ perspectives, is a key element that likely explains the diversity in healing rituals, narratives, and osteopathic techniques observed in clinical care. However, in the absence of a clear ethical professional framework, distinctions between pseudoscientific/nocebo narratives and narratives rooted in traditional/CAM, both of which are currently observed in the osteopathic profession, remain unclear. This problem needs to be urgently addressed by the osteopathic profession not because of our blurred professional identity but because we need to protect patients and make sure they *all* receive culturally sensitive, patient-centered, evidence-informed care.

## Figures and Tables

**Figure 1 healthcare-11-00010-f001:**
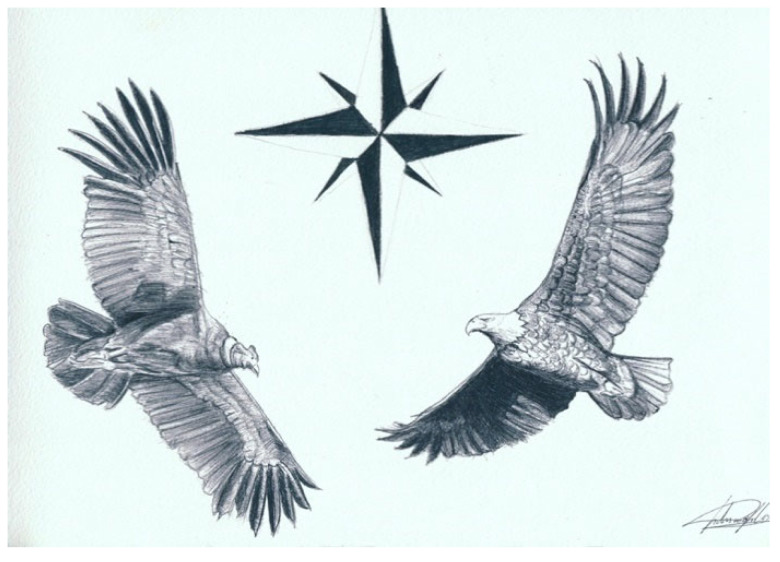
The Andean prophecy of the Eagle and the Condor flying together. Drawing by Gianluca Pompilio, Roma, Italy.

**Figure 2 healthcare-11-00010-f002:**
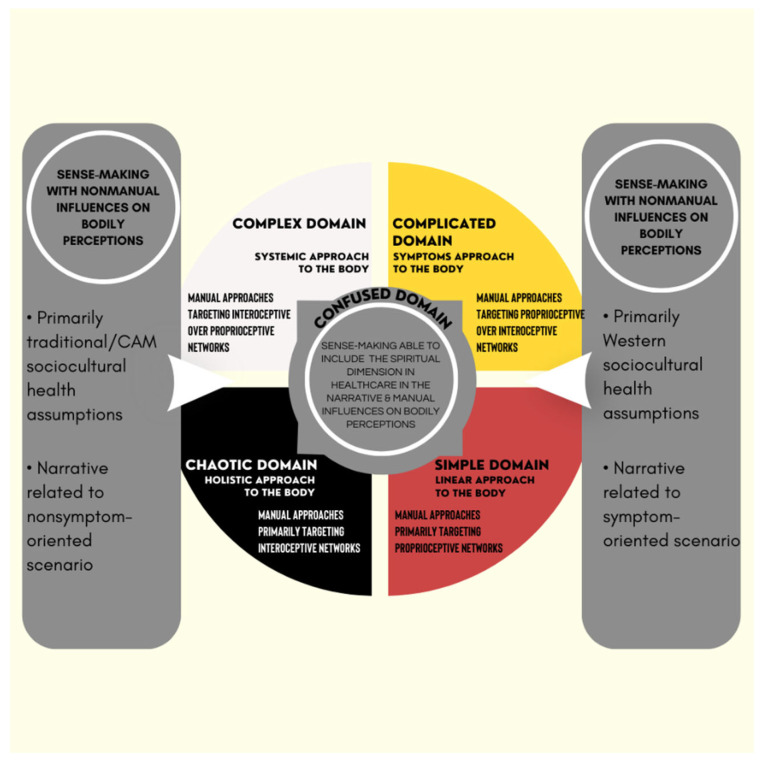
A culturally sensitive, patient-centered, evidence-informed approach using the Cynefin framework for osteopathic care (adapted from Zegarra-Parodi et al. [78]).

**Table 1 healthcare-11-00010-t001:** Professional considerations to promote culturally sensitive, patient-centered, evidence-informed osteopathic care among practitioners (adapted from Asnaani and Hofmann [66]).

Professional Skills		Consideration of Professional Limitations for Patient Values and Expectations Based on Different Sociocultural Health Assumptions	
		**Traditional-Minded/Condor Spirit-Based Healing Wisdom**	**Evidence-Minded/Eagle Scientific-Based Healing Wisdom**
#1	Adopt an evidence-informed practice similar to all other Western healthcare professionals to ensure safe and ethical care and optimize treatment outcomes.	If necessary, consider that ‘evidence’ also includes Western biomedical literature.	If necessary, consider that ‘evidence’ also includes medical anthropology literature.
#2	Be aware of the importance of respecting patients’ sociocultural health assumptions when they differ from practitioners’ assumptions.	If necessary, consider continuing professional development courses on Western ethical standards of care for regulated professions.	If necessary, consider continuing professional development courses in cross-cultural competencies to embrace the diversity of patients’ sociocultural health assumptions.
#3	Engage in self-education about specific cultural norms towards health and consult the literature for culture-specific treatment options.		
#4	Ensure adequate and effective training of practitioners in cross-cultural competency related to the existing diversity of sociocultural health assumptions in the Western environment.		
#5	Conduct a culturally informed but person-specific clinical assessment of the presenting problem.		
#6	Explore the patient’s perspective for seeking osteopathic care (nonsymptom-oriented vs. symptom-oriented scenarios) and the subsequent nature of the therapeutic alliance.	If necessary, consider nonsymptom-oriented scenarios in the complex, chaotic and confused domains of the Cynefin framework to guide osteopathic care.	If necessary, consider symptom-oriented scenarios in the simple and complicated domains of the Cynefin framework to guide osteopathic care.
#7	Identify and incorporate patients’ culturally related strengths and resources into treatment options.		
#8	Identify and utilize technique-specific and narrative-specific cultural modifications to meet patient values and expectations aligned with their sociocultural health assumptions.		

## Data Availability

Not applicable.

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
