# Peer review of "Historical Osteopathic Principles and Practices in Contemporary Care: An Anthropological Perspective to Foster Evidence-Informed and Culturally Sensitive Patient-Centered Care: A Commentary"

_healthcare, 2022, doi:10.3390/healthcare11010010_

Round 1
Reviewer 1 Report
Reviewer Comments
Thank you very much for the opportunity to review the manuscript submission entitled: Historical osteopathic principles and practices in contemporary care: an anthropological perspective to foster evidence-informed and culturally sensitive patient-centered care.
The current medical anthropological perspective on the legacy of traditional/CAM principles in historical OPP is offered to advance the osteopathic profession by proposing inclusive approaches for patients and promoting ethical, culturally sensitive, evidence-informed PCC in a Western secular environment; however, some limitations and constructive comments are pointed out below:
Specific comments
The text must be proofread in English to correct all grammatical errors.
Title and abstract:
· Include the study design in the title.
· What is the clinical significance? It should be mentioned in the conclusions.
· Include MeSH terms as keywords.
Introduction
· Describe the rationale for the review in the context of what is already known.
· Refere to similar study published” Vogel S, Zegarra-Parodi R. Relevance of historical osteopathic principles and practices in contemporary care: Another perspective from traditional/complementary and alternative medicine. International Journal of Osteopathic Medicine. 2022 Apr 21.
· Emphasize on the key question(s) identified for the essay topic.
Methods, discussion, and conclusion
· Eventhough it is an essay specify the process for identifying the literature search (eg, years considered, language, publication status, study design, and databases of coverage).
· Discuss: 1) research reviewed including fundamental or key findings, 2) limitations and/or quality of research reviewed, and 3) need for future research.
· Provide an overall interpretation of the essay in the context of clinical practice and implementation, or future research.
Reviewer 2 Report
First of all, I want to note that it has been a pleasure review your manuscript. I think this is an interesting topic, now that osteopathy is being labelled a pseudo-science
This study assesses the current challenges of evaluation of historical osteopathic principles and practices through the dominant Western biomedical lens taking into account an anthropological perspective of the non-Western traditional/complementary and alternative principles.
This essay, from my point of view, provides different approaches, with a wide and exhaustive bibliographic search with 93 references. The different parts into which the manuscript has been divided to facilitate the reader's understanding are appreciated.
In order to improve the quality of the manuscript. After reading in depth the manuscript, I would like to make some comments and ask the authors several questions about.
- - line 6: correct space in affiliation 1
- - Line 35. It would be better if you put the full name of Dr. Still and what the acronyms mean, although they are well known in the field of osteopathy but in other areas they may not be so well known and the reader may not understand them.
- - lines 120, 134, 148, 309,790. 791... Correct, please, the final of the sentence.
- - Perhaps it could be completed by commenting on some law or regulation that has been made in your country of origin or another country, which deals with the field of osteopathy and could be significant in this manuscript..
- the 610 line is not fully visible.
- - 651-654 is part of the figure legend or part of the text? I assume it is part of the text, so the font size will need to be changed.
- - References 15 and 68 have been slightly misconfigured.
- - In general, the text speaks of osteopathic doctors, but as it says in one of its sections, not only osteopaths can be doctors, in Spain, for example, physiotherapists also practice osteopathy.
Round 2
Reviewer 1 Report
The authors have addressed all the comments satisfactorily. The Manuscript has been improved significantly. It can be accepted for publication.